# Working Mechanisms of Nanoscale Light-Emitting Diodes Operating in Non-Electrical Contact and Non-Carrier Injection Mode: Modeling and Simulation

**DOI:** 10.3390/nano12060912

**Published:** 2022-03-10

**Authors:** Wenhao Li, Kun Wang, Junlong Li, Chaoxing Wu, Yongai Zhang, Xiongtu Zhou, Tailiang Guo

**Affiliations:** 1College of Physics and Information Engineering, Fuzhou University, Fuzhou 350108, China; 201127129@fzu.edu.cn (W.L.); 211110018@fzu.edu.cn (K.W.); 211120029@fzu.edu.cn (J.L.); gtl_fzu@hotmail.com (T.G.); 2Fujian Science & Technology Innovation Laboratory for Optoelectronic Information of China, Fuzhou 350108, China

**Keywords:** nano-LED, non-carrier injection, working mechanism, alternating current, simulation

## Abstract

Non-electrical contact and non-carrier injection (NEC&NCI) mode is an emerging driving mode for nanoscale light-emitting diodes (LEDs), aiming for applications in nano-pixel light-emitting displays (NLEDs). However, the working mechanism of nano-LED operating in NEC&NCI mode is not clear yet. In particular, the questions comes down to how the inherent holes and electrons in the LED can support sufficient radiation recombination, which lacks a direct physical picture. In this work, a finite element simulation was used to study the working process of the nano-LED operating in the NEC&NCI mode to explore the working mechanisms. The energy band variation, carrier concentration redistribution, emission rate, emission spectrum, and current-voltage characteristics are studied. Moreover, the effect of the thickness of insulating layer that plays a key role on device performance is demonstrated. We believe this work can provide simulation guidance for a follow-up study of NEC&NCI-LED.

## 1. Introduction

Closing the gap between the real world and display images to obtain a more realistic visual experience is the overall goal of display technology [1,2,3,4,5,6]. For this goal, nano-pixel light-emitting display (NLED) is an emerging display technology with ultrahigh pixel density higher than 10,000 pixel per inch [7]. Further reductions in pixel size to the nanoscale can enhance the sense of reality of display images and reduce the size of the display device. The key characteristic of NLED is that each pixel of NLED consists of multiple or even a single nanoscale light source. Thus, NLED is considered as the ultimate light source for light field displays, eye computer interface, and implantable displays (Figure 1) [7]. It is no doubt that the core component of NLED is the nanoscale light emitting device (nano-LED). Applying gallium nitride (GaN)-based LED technology to NLED is effective [8]. However, once the size of LED reduces to nanoscales, pixel-level driving is difficult, because the efficient electrical connection between the electrode array and the nano-LED array is challenging [9].

Recently, an operation mode for micro-LEDs/nano-LEDs, namely, non-electrical contact and non-carrier injection (NEC&NCI) mode, has been demonstrated [10,11,12,13]. It is believed that, by using NEC&NCI technology, the manufacturing difficulty of NLED can be effectively reduced. For LEDs operating in the NEC&NCI mode, the self-supporting LED chips composed of n-GaN/multi-quantum wells (MQWs)/p-GaN are sandwiched between insulting layers, and an alternating current (AC) electric field along the direction perpendicular to MQWs is applied. Obviously, no electrical contact exists between external electrodes and the LED chip. Therefore, the nano-LED operating in the NEC&NCI mode has an ultrasimple structure. For example, conventional GaN-LED requires a transparent contact layer, an upper p-electrode, a bottom n-electrode, and postprocessing for good contact between functional layers. Additional, poor ohmic contact and current crowding can lead to a decrease in device efficiency [14,15,16,17,18,19,20]. However, the transparent contact layer, p-electrode, and n-electrode are eliminated in the NEC&NCI operation mode, and the elaborate design of the energy band alignment of electron/hole injection layers, transport layers, and emissive layers can be eliminated. Thus, the device structure of nano-LEDs applied to NLED can be simplified, and manufacturing difficulty can be effectively reduced.

Due to the existence of insulating layers between the electrodes and the LED chip, external carriers cannot be injected into the LED chip, and only the inherent holes and electrons contribute to radiative recombination. Therefore, NEC&NCI-LEDs can only work under an AC electric field to obtain periodic electroluminescence (EL), which is different from conventional LEDs. For conventional LEDs, whether in direct current (DC) mode or in AC mode, holes are injected from p-GaN and the electrons are injected from n-GaN to achieve continuous EL [21,22,23,24,25,26,27,28,29,30]. Therefore, the working mechanisms and photoelectric characteristics of NEC&NCI-LED are different from the conventional injection mode. In our previous work, we have proposed a reasonable working mechanism mode for NEC&NCI-LEDs [10,11,12]. It is proposed that the forward field causes a diffusion of majority carriers and subsequent radiative recombination in MQWs. The reverse field drifts the carriers to their original state in preparation for the next EL process. However, the working mechanism of nano-LED in NEC&NCI mode is not clear yet. A direct physical picture of how the inherent holes and electrons provide sufficient radiative recombination under an AC electric field is lacking.

In this work, the finite model of a nano-LED operating in NEC&NCI mode is established, and the working mechanism is studied. The energy band variation and carrier concentration redistribution under AC field are quantitatively demonstrated, which can provide a clear physical image for the working mechanism of NEC&NCI-LED. The photoelectric characteristics, including emission rate, emission spectra, and current-voltage relationship, are presented. Moreover, the effect of insulating layer thickness that plays a key role on device performance is demonstrated. We believe this work can provide simulation guidance for a follow-up study of NEC&NCI-LED.

## 2. Model Details

Schematic structure of the nano-LED operating in NEC&NCI model is demonstrated in Figure 2a, where the nano-LED is sandwiched between two insulting layers. The external electric field generates an electric potential drop in the nano-LED, which would drive the inherent carriers to recombine at MQW [31]. Obviously, the device is equivalent to a resistor-capacitance (RC) parallel circuits [11,12]. As shown in Figure 2b, R_LED_ is the internal resistance of the nano-LED, and C_LED_ is the capacitor related to PN junction. C_x1_ and C_x2_ are equivalent external capacitors, which are related to the insulating layers on both sides.

The finite model of the nano-LED operating in the NEC&NCI mode is established, as shown in Figure 2c. In this model, Al_2_O_3_ layers are used as the insulator to separate the self-supporting nano-LED from external electrodes. The sinusoidal voltage is applied to external electrodes. It should be noted that the insulator is a key component affecting device performance, and many dielectric materials may be good choices for NEC&NCI-LEDs. In our previous work, the Al_2_O_3_ layer was used as the insulator. Thus, we choose Al_2_O_3_ as the insulating material in this mode to study the working mechanism of NEC&NCI-LEDs. The nanorod LED has a diameter of 500 nm with commercial LED epitaxial structure. The detailed LED epitaxial structure is demonstrated in Figure 2d and Table 1. The device structure from top to bottom includes the Al_2_O_3_ insulating layer (100 nm), p-GaN (200 nm), Al_0.15_Ga_0.85_N electron blocking layer (20 nm), GaN (12 nm), MQWs, n-GaN (2000 nm), and Al_2_O_3_ insulating layer (100 nm), respectively. The MQWs’ structure from top to bottom include In_0.15_Ga_0.85_N (3 nm)/GaN (12 nm) (Loop 2) and In_0.15_Ga_0.85_N (3 nm)/In_x_Ga_1-X_N (12 nm) (X = 0.01~0.05) (Loop 5), where Loop 2 means the same structure is repeated twice and Loop 5 means the same structure is repeated five times.

## 3. Results and Discussion

### 3.1. Typical Electrical and Optical Properties

Figure 3a shows the waveforms of the applied voltage, measured current, and emission rate in one cycle when a sinusoidal voltage with an amplitude of 50 V is applied. It can be found that the current is ahead of the voltage due to the capacitive characteristics of the device, which is consistent with experimental results [11]. A narrow EL pulse only occurs in the positive half cycle of the voltage, and the EL peak can be obtained when the voltage reaches its maximum value, which is also consistent with experimental results [11]. As well known, the redistribution of charge carriers in nano-LEDs will generate an induced field that is opposite to the direction of the external electric field. As a result, the induced electric field will shield the applied field and prevent the recombination of electrons and holes. Therefore, only a narrow EL pulse can be observed.

The relationships between the peak current and frequency *(i*-*f*) at different voltage amplitudes are presented in Figure 3b. In particular, the detailed current-voltage relationship at low frequencies is shown in the inset of Figure 3b. Consistent with experimental results, the current increases exponentially with the increase in frequency [11]. According to differential calculation results, the detailed change of the peak current can be observed more visually, as shown in Figure 3c. In the low frequency range (<1 MHz), the current remains basically unchanged. At frequencies above 1 MHz, the current increases obviously, and maximum increasing rates can be obtained at ~500 MHz (30 V), ~800 MHz (50 V), and ~1 GHz (70 V), respectively. These numerical calculation details can help explain important device characteristics, such as the optimal frequency point for the emission rate and power efficiency, which will be analyzed in the following section.

The peak current-peak voltage (*i*-*v*) curves of the device at different frequencies and the relevant fit curves are presented in Figure 3d. It is interesting that the peak current is almost linear to the applied voltage within an allowable range of linear fitting error, which is consistent with the experimental results (inset of Figure 3d). Moreover, the peak current is sensitive to the driving frequency. The current increases with the increase in frequency at the same voltage. This possible reason is that NEC&NCI-LED can be equivalent with a series circuit of an LED and two capacitors, and the capacitive reactance of the device increases with the frequency of the applied voltage.

The EL spectra at different frequencies are calculated, as shown in Figure 3e. When the frequency increased from 5 MHz to 500 MHz, there is a blue-shift in the EL spectrum, which is consistent with experimental results. The blue-shift phenomenon is caused by the quantum-confined Stark effect [32,33]. It is well known that the polarized electric field formed by the polarized charges tilts the energy band of the MQWs; thus, when the number of electrons injected into MQW is relatively small, the emission wavelength is longer. The increase in voltage frequency leads to an increasing number of injected electrons. The electrons can shield the polarized electric field and weaken the quantum-confined Stark effect. Therefore, the ground state in the well rises, which shifts the LED peak wavelength toward shorter wavelengths. Additionally, full width at the half maximum (FWHM) of the EL spectra also increases with frequency, as shown in Figure 3f. This is because the increase in the number of electrons in MQW leads to an increase in the probability of radiative recombination at each location of the MQWs, thereby broadening the EL spectrum.

### 3.2. The Mechanism of Carrier Transport

Due to the inevitable difference between the parameters of the simulation model and the parameters of the actual device, there is a certain difference between experimental data and simulation data. However, the trend of the simulation results is consistent with experimental results. Therefore, it is expected that the simulation model can be used to more clearly understand the dynamic change process of the device energy band and carrier concentration, which is of great value for revealing the working mechanisms of the device and further optimizing the device’s performance. First, we investigate the energy band variation of the device. Because there is no electrical contact between external electrodes and the LED chip, the energy band of the NEC&NCI-LED is different from that of the LED operating in traditional mode. The energy band change in one operation cycle is demonstrated, as shown in Figure 4a. Significant energy bending in the MQWs region and device terminals (including p-GaN and n-GaN terminals) can be observed due to the redistribution of constant inherent carriers [34,35]. The energy bending in the device terminals means that an induced electric field opposite to the applied electric field is generated. In particularly, parameters related to the length of energy change (Δ*L_EB_*) and the value of energy change (Δ*E_EB_*) are proposed to quantitatively characterize energy bending, as shown in bottom right corner of Figure 4a.

During the increasing process of forward voltage (process I in Figure 4b), the capacitors of C_X1_ and C_X2_ are charging. The holes in the p-GaN region and the electrons in the n-GaN region diffuse into MQW, respectively. In this case, depletion regions are formed on both sides of the semiconductor (Δ*E_EB_* > 0). Therefore, energy bending in the p/n GaN terminals can be observed. The values of Δ*E_EB_* and Δ*L_EB_* reach the maximum at the peak voltage, as shown in Figure 4d,e. Quantitatively, the maximum values of Δ*E_EB-p_* and Δ*L_EB-p_* in the p-GaN terminal are 12.504 eV and 148.4 nm, respectively. The maximum values of Δ*E_EB-n_* and Δ*L_EB-n_* in the n-GaN terminal are 1.705 eV and 24.4 nm, respectively. It should be noted that because the doping concentration of p-GaN is lower than that of n-GaN, more obvious energy bending can be observed in the p-GaN terminal. Since the PN junction is in a forward-biased state, the change in Δ*E_EB-MQW_* is relatively small, as shown in Figure 4c. The maximum value of Δ*E_EB-MQW_* is only 0.259 eV.

As the forward voltage is decreasing (process II in Figure 4b), the induced electric filed is stronger than the applied field. Thus, the inherent carriers are moving to both terminals under the induced electric field to fill the depletion regions, and Δ*E*_EB_ decreases, which corresponds to the discharging process of C_X1_ and C_X2_. It should be noted that when the forward voltage reduces to 0 V (*t* = 50 ns), the state of the energy band is different from the initial state, as shown in Figure 4d,e. The reason is that the carrier concentration cannot completely restore the initial state due to the unidirectional characteristics of LED. Quantitatively, when the forward voltage reduces to 0 V (*t* = 50 ns), the values of Δ*E_EB-p_* and Δ*L_EB-p_* in the p-GaN terminal are 1.73 eV and 73.917 nm, respectively. The values of Δ*E_EB-n_* and Δ*L_EB-n_* in the n-GaN terminal are 0.2392 eV and 16.3 nm, respectively. The value of Δ*E_EB-MQW_* is 11.6598 eV.

In the process of increasing the reverse voltage (process III in Figure 4b), the direction of the induced field is the same as that of the applied field. Thus, the moving behavior of carriers is similar to process II. Finally, the depletion regions in the p/n-GaN terminals disappeared (Δ*E_EB_* = 0) and the accumulation layer is formed (Δ*E_EB_* < 0), as shown in the red regions in Figure 4d,e. During the accumulation period, the maximum values of Δ*L_EB-p_* and Δ*L_EB-n_* are 24.763 nm and 12.4 nm, respectively. The minimum values of Δ*E_EB-p_* and Δ*E_EB-n_* are −0.075 eV and −0.0837 eV, respectively. Since the PN junction is in a reverse biased state, the change in Δ*E_EB-MQW_* is relatively large. The maximum value of Δ*E_EB-MQW_* is 24.1722 eV, as shown in Figure 4c.

As the reverse voltage decreases (process IV in Figure 4b), the induced electric filed is stronger than the applied field. Thus, the carriers in the accumulation layers are moving far away from the terminals, which corresponds to the discharging process of C_X1_ and C_X2_. As a result, the accumulation regions of the p/n GaN terminals disappeared (Δ*E_EB_* = 0) and the depletion regions are reformed (Δ*E_EB_* > 0). When the reverse voltage reduces to 0 V, the Δ*E_EB-p_* and Δ*L_EB-p_* in the p-GaN terminal are 1.7289 eV and 71.516 nm, respectively. The Δ*E_EB-n_* and Δ*L_EB-n_* in the n-GaN terminal are 0.239 eV and 23.3 nm, respectively. The values of Δ*E_EB-MQW_* is 11.6538 eV. As shown in Figure 4d,e, at the end of one cycle (*t* = 100 ns), Δ*L_EB-n_* and Δ*L_EB-p_* are non-zero, which is different from the initial state (*t* = 0 ns). This means that when the reverse voltage reduces to 0 V, there is remarkable energy bending in the p/n-GaN terminals. The reason is that the carrier concentration cannot completely restore the initial state because of the unidirectional characteristics of the LED.

The dynamic change of the energy band reveals the state of inherent carriers in NEC&NCI-LED. To have a clearer understanding of the working status of the device, the change in carrier concentration within one cycle is further studied, as shown in Figure 5. It can be found that the carrier concentration changes dramatically in the terminals, which is completely different from traditional LED [36,37]. Similarly to the analysis of change in energy band, the parameters related to the length of the majority carrier concentration variation is defined as Δ*L*_cc_, as shown in Figure 5a.

During the increasing process of the forward voltage (process I in Figure 5b), holes in the p-GaN region and electrons in the n-GaN region diffuse into MQW, respectively, as schematically presented in Figure 6a. Thus, the hole concentration in the p-GaN terminal and the electron concentration in the n-GaN terminal are reduced, and the depletion regions are formed on both terminals. The concentration of hole in the p-GaN terminal (*p_p_*) and electron in the n-GaN terminal (*n_n_*) reach the minimum (*p_p_* = 6.1804 × 10^−64^/cm^3^, *n_n_* = 2.4492 × 10^−11^/cm^3^) when the voltage reaches the peak value, as shown in Figure 5c,d. The maximum Δ*L_cc-p_* in the p-GaN terminal and Δ*L_cc-n_* in the n-GaN terminal are 139.32 nm and 29.7 nm, respectively. In this case, the minority carrier concentrations in the p-GaN terminal (*n_p_*) and the n-GaN terminal (*p_n_*) are 2520.6/cm^3^ and 2.6968 × 10^11^/cm^3^, respectively. The maximum value of Δ*L_cc-p_* is 70 percent of the length of p-GaN (Figure 5c), and the maximum value of Δ*L_cc-n_* is 1.5 percent of the length of n-GaN (Figure 5d). This means that the holes in p-GaN are more sensitive to the applied voltage. Thus, in the design of NEC&NCI-LEDs, increasing the length of p-GaN and reducing the length of n-GaN can help improve the utilization rate of carriers.

As the forward voltage decreases (process II in Figure 5b), the carriers move to both terminals to fill the depletion regions under the induced electric field, as schematically presented in Figure 6b. Therefore, *p_p_* and *n_n_* increased. At this time, *p_n_* and *n_p_* are reduced. It should be noted that when the voltage reduces to 0 V (*t* = 50 ns), the carrier concentrations are different from the initial state, which is similar to the variation of the energy band (Figure 4). When the forward voltage reduces to 0 V, *p_p_* and *n_n_* are 1.2631 × 10^−12^/cm^3^ and 3.8679 × 10^14^/cm^3^, respectively. The Δ*L_cc-p_* in the p-GaN terminal and Δ*L_cc-n_* in the n-GaN terminal are 57.9 nm and 17.3 nm, respectively. The values of *n_p_* and *p_n_* are 1977.4/cm^3^ and 3.2871 × 10^11^/cm^3^, respectively. When the forward voltage reduces to 0 V, the depletion region still exists. In order to restore the device to the initial state faster, the addition of electron and hole transport layers can be considered.

In the process of increasing the reverse voltage (process III in Figure 5b), the moving behavior of carriers is similar to the process II. *p_p_* and the *n_n_* increased, as schematically demonstrated in Figure 6c. Finally, the depletion region of the p/n-GaN terminals disappear (The *p_p_* is 7 × 10^17^/cm^3^ and the *n_n_* is 5 × 10^18^/cm^3^) and the accumulation layer is formed, as shown in the red regions in Figure 5c,d. The maximum *p_p_* and *n_n_* are 1.3706 × 10^19^/cm^3^ and 1.9001 × 10^19^/cm^3^ respectively. The maximum Δ*L_cc-p_* in the p-GaN terminal and Δ*L_cc-n_* in the n-GaN terminal are 20.923 nm and 13.4 nm, respectively. The minimum *n_p_* and *p_n_* are 1.2791 × 10^−32^ cm^3^ and 60.338/cm^3^, respectively. The accumulation process of carriers is the most important in process III. In order to provide enough carriers to participate in light emission in the next cycle, the accumulation time can be increased to make carriers move toward p/n-GaN terminals as much as possible.

As the reverse voltage decreases (process IV in Figure 5b), the carriers in the accumulation layers move far away from the terminals under the induced electric field, as shown in Figure 6d. The hole concentrations in the p-GaN terminal and the electron concentrations in the n-GaN terminal are reduced. As a result, the accumulation regions of the p/n-GaN terminals disappeared and depletion regions formed. When the reverse voltage reduces to 0 V (*t* = 100 ns), *p_p_* and *n_n_* are 1.3226 × 10^−12^/cm^3^ and 3.8917 × 10^14^/cm^3^, respectively. The Δ*L_cc-p_* in the p-GaN terminal and Δ*L_cc-n_* in the n-GaN terminal are 81.347 nm and 17.3 nm, respectively. *n_p_* and *p_n_* are 8.0056 × 10^−27^/cm^3^ and 0.13865/cm^3^, respectively. Within one cycle, the carrier concentrations at the initial state (*t* = 0 ns) are different from that at the final state (*t* = 100 ns). The results show that due to the unidirectional characteristics of the LED, the carrier concentration cannot fully recover the initial state, which is consistent with the energy bending analysis shown in Figure 4. Therefore, in order to improve device performance, the device structure should be optimized so that the final state is consistent with the initial state.

The change in carrier concentration reveals that the working mechanism of NEC&NCI-LED is completely different from traditional LED. As is well known, the number of carriers in the MQW affects EL intensity [38]. Thus, the change in the carrier concentration in the MQW is furthered studied, as shown in Figure 7a–d. In the positive half cycle, the concentration of electrons and holes in the MQW increases initially, followed by a decrease, as shown in Figure 7a,b. During the increasing process of the forward voltage (0-25 ns), the holes in the p-GaN region and the electrons in the n-GaN region diffuse into MQW, respectively. The orders of magnitude of electrons and holes concentration are 10^18^/cm^3^ and 10^17^/cm^3^, respectively. Thus, EL can be observed.

As the forward voltage decreases (25–50 ns), the induced electric field is formed and shields the applied field. As a result, the value of the carrier concentration in MQWs slightly decreased. When voltage reduces to 0 V, the orders of magnitude of electrons and holes concentration are 10^15^/cm^3^ and 10^7^/cm^3^, respectively. As the reverse voltage is increasing (50–75 ns), the majority carrier concentration in MQW further decreases until the terminal carrier concentration reaches the maximum. At this time, the orders of magnitude of electrons and holes concentration are 10^12^/cm^3^ and 10^2^/cm^3^, respectively. In the remaining period of time (75–100 ns), the concentration of electrons and holes remain basically unchanged, as shown in Figure 7c,d.

### 3.3. Frequency Response Characteristics

As well known, the frequency of the applied voltage is a critical parameter that would impact the luminous characteristics of NEC&NCI-LED [39,40]. This is because the carrier concentration in MQWs will change with the driving frequency. In order to explore the influence of frequencies on the device performance, the variation of carrier concentration in the MQWs with different frequencies is presented, as shown in Figure 8a. From p-GaN side to n-GaN side, the QWs are defined as QW_1_, QW_2_, QW_3_, QW_4_, QW_5_, QW_6_, and QW_7_, respectively. The electron and hole concentrations in QW_1_ to QW_7_ are defined as n_1_ to n_7_ and p_1_ to p_7_, respectively. Considering that there are seven QWs in the MQWs, revealing the exact carrier concentration in each QW is important. On the one hand, it is helpful to analyze the working mechanism of the device by exploring the variation of the carrier concentration in each QW. On the other hand, analyzing the changes in the carrier concentration at different frequencies can help to find potential methods to improve luminescence intensity. Generally, the carrier concentration in MQWs increases initially and then decreases. At ~1 MHz, the hole concentrations in QW_1_ to QW_7_ are 4.6179 × 10^11^/cm^3^, 6.0531 × 10^13^/cm^3^, 5.7997 × 10^15^/cm^3^, 1.1941 × 10^17^/cm^3^, 5.3281 × 10^17^/cm^3^, 9.1368 × 10^17^/cm^3^, and 3.3103 × 10^18^/cm^3^, respectively. Moreover, the electron concentrations in QW_1_ to QW_7_ are 6.5236 × 10^16^/cm^3^, 6.7363 × 10^16^/cm^3^, 8.9176 × 10^16^/cm^3^, 1.6809 × 10^17^/cm^3^, 3.8208 × 10^17^/cm^3^, 5.3344 × 10^17^/cm^3^, and 1.7899 × 10^17^/cm^3^, respectively. The concentrations of electrons and holes in the QW_4_, QW_5_, and QW_6_ are very close at ~1 MHz (the order of magnitude is 10^17^). At ~800 MHz, the summation of electrons in the MQW is 1.31 × 10^19^/cm^3^ and the summation of holes in the MQW is 1.98 × 10^19^/cm^3^, which reaches the maximum. NEC&NCI-LED is equivalent to a series circuit of an LED and two capacitors. Thus, the charging current increases with the frequency and the carrier concentration in MQW increases initially. However, the moving distance of carriers will change with frequencies. When frequency is low, the carriers in the device have sufficient time to diffuse into MQW due to the relative small lifetime of the carriers. Once the frequency is high enough, the movement of carriers cannot keep up with the change of applied electric field due to the relative small lifetime of carriers. As a result, the number of carriers in the MQW is reduced.

The emission rate and the power efficiency at different driving frequencies are presented in Figure 8c. The emission rate and power efficiency also show a trend of increases initially and then decreases, which is consistent with experimental results [11]. The optimal frequency point of emission rate is ~800 MHz, which corresponds to the frequency point where the summation of the carrier in MQWs reaches the maximum. Thus, the probability of radiative recombination is maximized. This phenomenon can be directly reflected by the carrier concentration of MQW, as shown in Figure 8a. The optimal frequency point of power efficiency is ~1 MHz. The concentration of electrons and holes in the QW_4_, QW_5_, and QW_6_ are very close at ~1 MHz (the order of magnitude is 10^17^) and the number of QWs with the same order of magnitude of carriers reaches maximum, as shown in Figure 8b. Therefore, the carrier utilization rate is relatively high. On the other hand, the current remains basically unchanged in the low frequency range (<1 MHz). As a result, a maximal power efficiency can be obtained at ~1 MHz. For NEC&NCI-LEDs, only inherent holes and electrons contribute to radiative recombination. It should be noted that the number of carriers for radiative recombination is much less than that of conventional LEDs, which results in the low output luminous power. On the other hand, due to the existence of insulating layers, the working voltage of NEC&NCI-LEDs is greater than that of conventional LEDs, and only a small portion of the voltage is applied to the LED. Thus, the power efficiency of NEC&NCI-LEDs is low.

The thicknesses of the insulator are critical parameters that can impact the luminous characteristics of NEC&NCI-LED [12]. In order to explore the influence of insulator thicknesses on the device, the emission rates of the devices with different insulator thicknesses are presented, as shown in Figure 8d. As thickness increases, the emission rate shows a downward trend. The value of the emission rate is reduced by 11 times as the insulator thicknesses increase from 50 nm to 1000 nm. The value of the current directly reflects the concentration of carriers for movement. This is because the impedance of the equivalent capacitors (C_x1_ and C_x2_) can increase with the increase in insulator thicknesses. From the simulation results, the device performance is better when the insulating layer is thinner. However, an insulating layer that is too thin can result in device breakdown. In order to obtain the optimized structure of a real device, it is necessary to combine simulations and experiments.

## 4. Conclusions

In summary, we using finite element simulation to study the working mechanism of the nano-LED operating in the NEC&NCI mode. The energy band, carrier concentration redistribution, emission rate, emission spectrum, and current-voltage characteristics are studied. It is found that the change of the energy band structures and carrier concentrations in terminals of NEC&NCI-LED is different from that of conventional LEDs. This is because there is no injection of external carriers to supplement the carriers flowing away from the p/n GaN regions. The performances of the nano-LED operating in the NEC&NCI mode are highly sensitive to frequencies of the applied voltage. We also demonstrate that the insulator thickness is a key parameter impact on the luminous properties of NEC&NCI-LEDs. We hope that this work can provide a simulation guidance for the follow-up study of NEC&NCI-LED.

## Figures and Tables

**Figure 1 nanomaterials-12-00912-f001:**
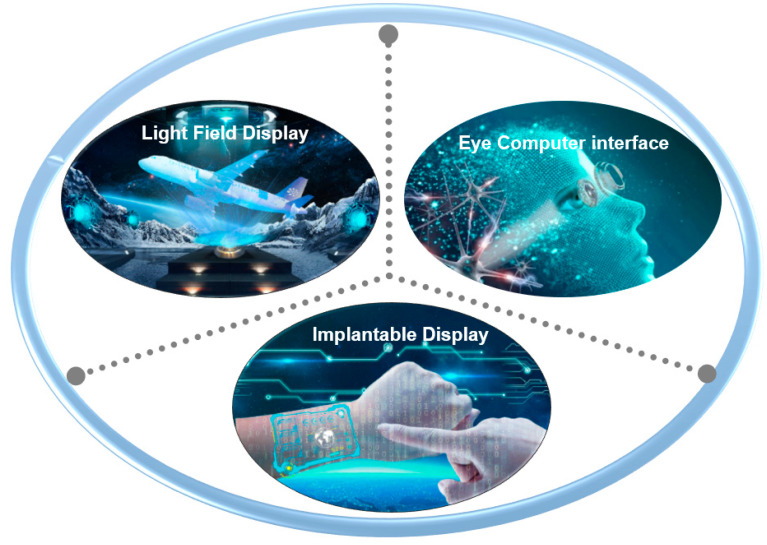
Potential applications of NLED.

**Figure 2 nanomaterials-12-00912-f002:**
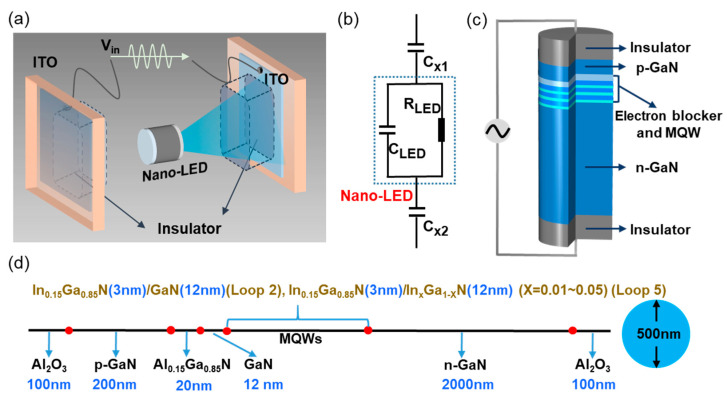
Schematic diagram of the nano-LED operating in NEC&NCI mode and the finite model for simulation. (**a**) Schematic of the nano-LED operating in the NEC&NCI mode. (**b**) Equivalent circuit of the device. (**c**) Finite model for simulation. (**d**) Structural parameters of the model.

**Figure 3 nanomaterials-12-00912-f003:**
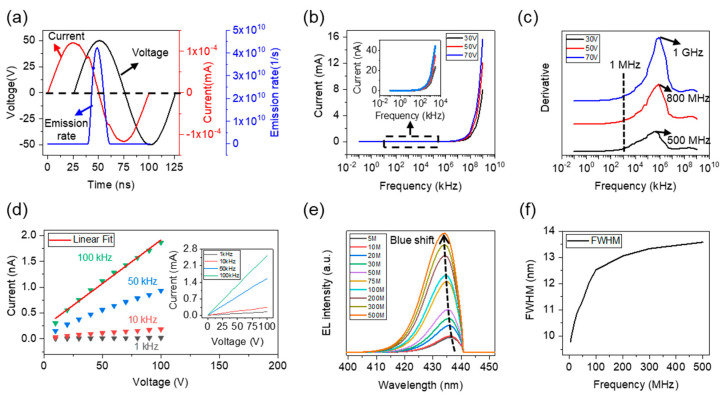
Electrical and optical properties of the nano-LED operating in the NEC&NCI mode. (**a**) Waveform of the applied sinusoidal voltage, measured current, and emission rate of the device in one AC voltage cycle. (**b**) Relationship between the peak current and frequency at different voltage amplitudes. (**c**) Derivative of the peak current. (**d**) Peak current-peak voltage curves at different frequencies. Inset: experimental *i*-*v* curves. (**e**) EL spectra at different frequencies. (**f**) Relationship between FWHM and the frequency.

**Figure 4 nanomaterials-12-00912-f004:**
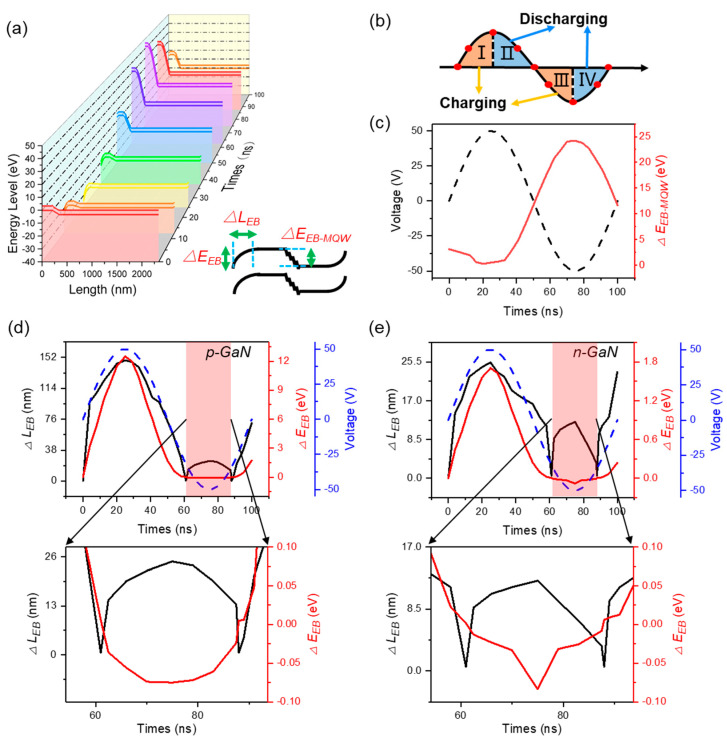
Energy band change of nano-LED operating in NEC&NCI mode. (**a**) Energy band in a voltage cycle. Bottom right corner: definition of Δ*L_EB_* and Δ*E_EB_*. (**b**) Schematic charging and discharging process. (**c**) Energy band in the MQWs region. (**d**) Energy band in the p-GaN terminal. (**e**) Energy band in the n-GaN terminal.

**Figure 5 nanomaterials-12-00912-f005:**
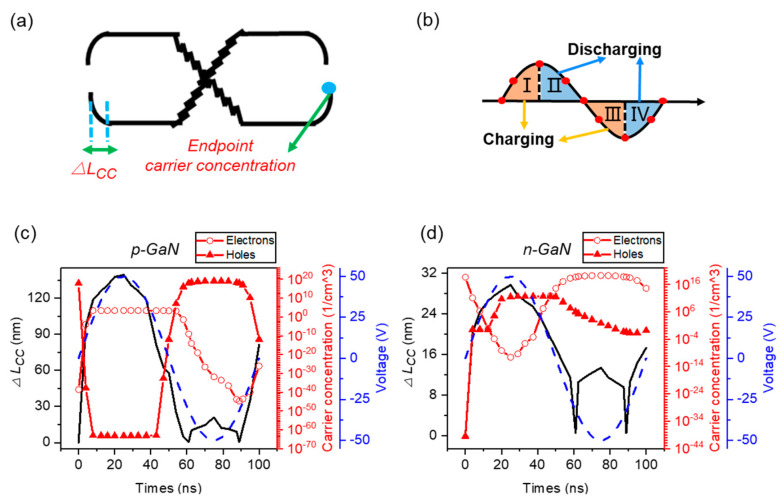
Change of the carrier concentration in p-GaN/n-GaN terminals. (**a**) Definition of Δ*L*_cc_. (**b**) Schematic charging and discharging process of the equivalent capacitance. (**c**) Carrier concentration in the p-GaN terminal. (**d**) Carrier concentration in the n-GaN terminal.

**Figure 6 nanomaterials-12-00912-f006:**
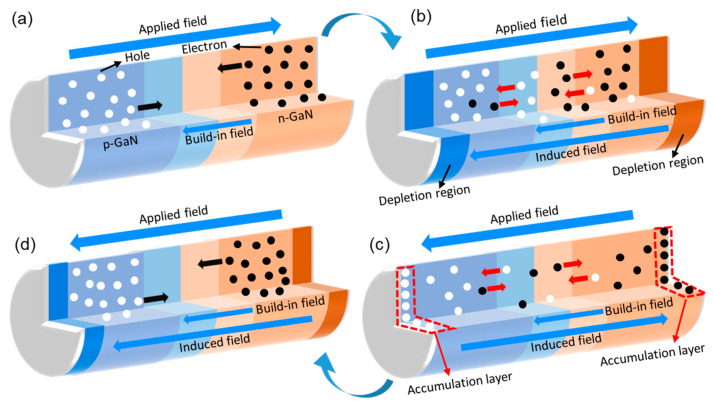
Schematic showing the operation mechanisms of the nano-LED in NEC&NCI mode. (**a**) Radiative recombination in the MQW region under forward bias. (**b**) Formation of an induced electric field that shields the external field. (**c**) Carriers accumulate in the p-GaN/n-GaN terminals of the nano-LED under reversed bias. (**d**) Movement of the accumulated carriers.

**Figure 7 nanomaterials-12-00912-f007:**
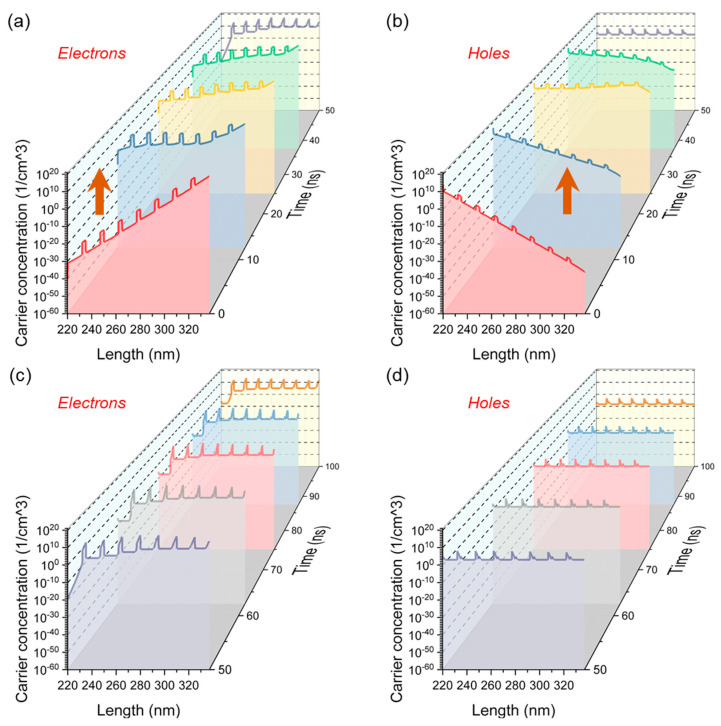
Variation of the carrier concentration in MQW region. (**a**) Electron concentration in the positive half cycle. (**b**) Hole concentration in the positive half cycle. (**c**) Electron concentration in the negative half cycle. (**d**) Hole concentration in the negative half cycle.

**Figure 8 nanomaterials-12-00912-f008:**
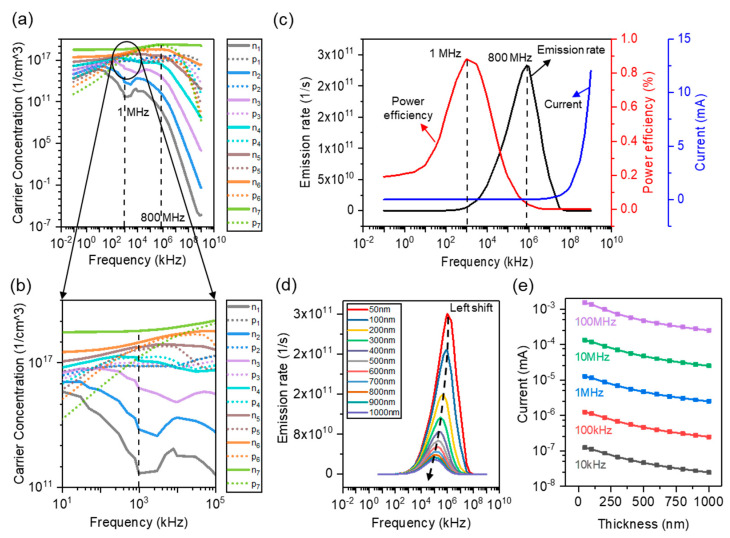
Frequency response characteristics of the device. (**a**) Carrier concentration in the MQWs at different frequencies. (**b**) Carrier concentration in the MQWs at 1 MHz. (**c**) Emission rate and the power efficiency at different frequencies. (**d**) Emission rate with different insulator thicknesses at different frequencies. (**e**) Peak current with different insulator thicknesses at different frequencies.

**Table 1 nanomaterials-12-00912-t001:** Nano-LED epitaxial structure parameters.

Materials	Band Gap /V	Electron Affinity /V	Electron Mobility/(cm^2^/(V × s))	Hole Mobility/(cm^2^/(V × s))	Doping Concentration/(1/cm^3^)
p-GaN	3.4	4.1	1000	350	7 × 10^17^
n-GaN	3.4	4.1	1000	350	5 × 10^18^
In_0.01_Ga_0.99_N	3.35884	4.12881	1000	350	/
In_0.02_Ga_0.98_N	3.31797	4.15742	1000	350	/
In_0.03_Ga_0.97_N	3.27739	4.18583	1000	350	/
In_0.04_Ga_0.96_N	3.23709	4.21404	1000	350	/
In_0.05_Ga_0.95_N	3.19708	4.24205	1000	350	/
In_0.15_Ga_0.85_N	2.81268	4.51113	1000	350	/
Al_0.15_Ga_0.85_N	3.693	3.575	1000	350	7 × 10^17^

## Data Availability

Not applicable.

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
