# Peer review of "Working Mechanisms of Nanoscale Light-Emitting Diodes Operating in Non-Electrical Contact and Non-Carrier Injection Mode: Modeling and Simulation"

_nanomaterials, 2022, doi:10.3390/nano12060912_

Round 1

Reviewer 1 Report

This submission paper discussed the theoretical non-electrical contact and non-carrier injection mode for nanoscale LEDs. This LED operation mode is a very interesting technology for ultrahigh dense pixel LED display. Also, it is very challenging. However, the submitted paper needs to discuss more details and clarify. Please see the comments and questions below.  

1. Why is the power efficiency to be so low?  Please explain the detail clearly.

2. Why did the authors use an Al2O3 for the insulator? The reviewer thinks the insulator material is also a key factor for device performance. Do you have any idea to improve the efficiency using proper materials?

3. In Fig. 8 (a), the carrier concentrations of hole and electron were different behavior when the LEDs operated with different frequencies. Especially, the differences of the carrier concentrations are in the low and high frequencies. Also, why did the carrier concentrations drop with high frequency? Please explain in more detail about it.

Reviewer 2 Report

In this work, finite element simulation is used to study the working process of the nano-LED operating in NEC&NCI mode to explore the working mechanism. The authors investigated the energy band variation, carrier concentration redistribution, emission rate, emission spectrum, and current-voltage characteristics. This work will be of interest to other researchers in scientific and engineering community of GaN-based LEDs. While the paper has some interesting constituents and seems technically sound, it also seems to be missing some details and would benefit from a more elaborate analysis and description of the results. In more detail, I consider the following points to require elaboration:

1) The authors write: “Because there is no external carriers injected into the LED chip, the periodical and stable electroluminescence (EL) can only be obtained under AC electric field…” Here, the authors should give the full name of AC.

2) The authors write: “For example, conventional GaN-LED requires a transparent contact layer, an upper p-electrode, a bottom n-electrode, and a post-processing for well contact between functional layers [14,15].” and “For traditional LED, regardless of direct current (DC) mode or AC mode, the holes injected from the p-GaN 58 and the electrons injected from the n-GaN for continuous electroluminescence [16-25].” Here, the general reference list seems a bit thin, considering the rapid evolution in the field. To give the readers a much broader view, several important references related to GaN-based LEDs, such as Applied Physics Letters 118, 182102 (2021); Optics Express 27, A669 (2019); Optics Express 25, 26615 (2017); Japanese Journal of Applied Physics 56, 111001 (2017), etc., should be included, so that the readers can be clear about the state-of-the-art of this topic.

3) In Fig.3e, “capacitive reactance” should be revised be “Capacitive Reactance”

4) The author should give FWHM of EL spectra?

5) The authors write: The number of carriers in the MQW affects the electroluminescence intensity. Here, EL should be used instead of electroluminescence.

6) The authors argue: there is a significant blue shift from 5MHz to 500MHz. What is the underlying physical mechanism for this blue shift? The detailed explanation should be provided.

7) In Fig.4d and e, “votage” should be corrected to be “voltage”

8) There are some grammatical errors in the manuscript, although most of them do not obscure the understanding of the technical contents. However, I believe that the paper should be proof-read for English before it is submitted. For example:

--“Non-electrical contact and non-carrier injection (NEC&NCI) mode is an emerging driving 11 mode for nanoscale light-emitting diodes (LEDs)…” should be corrected to be “Non-electrical contact and non-carrier injection (NEC&NCI) mode are an emerging driving 11 mode for nanoscale light-emitting diodes (LEDs)…”

--“It is believe that by using the NEC&NCI technology…” should be corrected to be “It is believed that by using the NEC&NCI technology…”

-- “…and the working mechanisms of the device is studied” should be corrected to be “…and the working mechanisms of the device are studied”

-- “the current increase exponentially” should be corrected to be “the current increases exponentially”

-- “The values of ΔEEB-MQW is 11.6598 eV” should be corrected to be “The value of ΔEEB-MQW is 11.6598 eV”

--“the depletion region in the p/n GaN terminals are disappeared” should be corrected to be “the depletion region in the p/n GaN terminals is disappeared”

--“the hole concentration in the p-GaN terminal and the electron concentration in the n-GaN terminal is reduced” should be corrected to be “the hole concentration in the p-GaN terminal and the electron concentration in the n-GaN terminal are reduced”

--“the pn and np is reduced” should be corrected to be “the pn and np are reduced”

--“The carrier concentration variation in the terminals reveal that the working mechanisms of NEC&NCI-LED is completely different from traditional LED.” should be corrected to be “The carrier concentration variation in the terminals reveals that the working mechanism of NEC&NCI-LED is completely different from traditional LED.”

Reviewer 3 Report

Nano-pixel NLED, where nano-LED is the core component, is an emerging technology the working process of which is not completely understood yet. This paper reports on numerical calculation using finite model of a nano-LED operating in non-electrical contact and non-carrier injection mode, in order to study the working mechanisms involved in this kind of device. Modelling and different simulations have been carried on the variation of the energy band and carrier concentration redistribution under AC field as well as some of the main photoelectric characteristic have been reported.

Many aspects of the working mechanisms are treated in this paper but more discussion about why are those so important and which configuration is the most promising, is still missing in my opinion. It is clear that periodical and stable electroluminescence can only be obtained under AC electrical field, but it is not mentioned which is its best configuration to reach best performances. In conclusion I do not think the authors are providing a clear physical image of the working mechanisms of the NEC&NCI LED as mentioned at the end of the introduction.

I have also the following points that I would like to bring up:

Introduction.

R39: it is believed;

R57-59: this sentence is not very clear.

R85: The external capacitors (Cx1 and Cx2) are related to the insulating layers on both sides or The external capacitors (Cx1 and Cx2) related to the insulating layers are on both sides;

Fig.2: I guess the text in brace refers to the electron blocker MQW, if so, wouldn't be clearer mention it just before the list of the element (yellow writing). What are Loop2 and loop5 referring to? By the way the parenthesis of loop2 and loop5 have different colors;

R97: N is missing at the end of Al0.15Ga0.85;

Result and Discussion

In general I do see results in this section but very little discussion. Why are all these numbers important and why the authors care to show them. Are those promising for this technology, should they be improved, do they need more investigation?

Fig3: from the derivative looks like the maximum of the slope in the current is, according to the dash line, around 1GHz (not 500 MHz as written in the plot). Of course this is not visible in the plot of the current, maybe a zoom in of that part of the curve could help. It is not clear from the text why there is an inset in the range between 10^-2 and 10^4 kHz. In general larger figures would help;

R127: (<1 MHz);

R128: 1 MHz to 10 GHz;

R129: from the dash line in fig 3 look like 1 GHz instead of 500 MHz;

R130: 0.5 THz;

R135: actually fig 3d shows IV at 1 MHz not at 10 MHz;

R136: “almost linearly”, does not mean to much. Try to make a linear fit and a polynomial fit and see if the difference is such that can suggest this kind of approximation;

R137-140: this part is not clear. Why is the linearity is so important and any guess why it is not as good as the experimental results. In general making so many times a comparison with experimental data, without showing them (only point to the citation) it is not fair in my opinion. For the clarity of the paper and its completeness, I think simulation and experimental data should come together when applicable.

Moreover why are different IVs as a function of the driving frequency taken, and what we can learn from it?

R141: again, “almost unchanged” does not mean too much to my opinion, just give a number suggesting the relative difference. It is hard to see from the graph (a grid would help) but I do see, for the purple curve (100 MHz) a capacitive reactance that goes from ~2e7 up to ~4e7 thus doubling its value. Is this "almost unchanged"?

R150: this statement cannot be verified from the previous section because there is no direct comparison between simulation and experimental results, there is only a recursive pointing to the reference [11];

R160: Due to;

R201: 24.763 nm is maximum for DeltaL_EB but -0.075 eV is a minimum for DeltaE_EB. Same in R202;

Fig5: I repeat myself, I see a lot of numbers but I would like to know the real meaning of those, are they ok and if yes why, are they better than normal LED or just the same? is that ok or not? I think this paper aim to be for general audience since it is not submitted to a conference proceeding where everyone know what you are talking about. At any rate the result of a research needs to be highlighted with more argument and discussion;

R248-249: I see only carrier concentration (red line), voltage (blue line) and carrier contraction variation (black line) in Fig 5g and 5f. I cannot compare the energy state between initial state and V=0, neither with fig.4;

R250-254: what do these number suggest? Discussion is needed to understand the potentiality of this technology;

R299: as well known;

R306 and Fig8: “The concentration of electrons and holes in the QW are approximately the same at ~1 MHz…”,  I think they differ from each other quite a lot. I do not see how the concentration of electrons and holes could be considered approximately equal, if I read the plot right: n1 is ~10^12, n2 is ~10^14, n3 is ~10^15, n4 and n5 are ~10^17, n6 is ~10^18 and n7 is ~50x10^18 @ 1 MHz.

In Fig8a The dashed line for 500 MHz is settled between 10^5 and 2x10^5 kHz, and 1MHz does not have a line to be followed. In Fig8c the dashed line corresponding to 500 MHz is again settled wrong, this time at 10^6 kHz. Include what n1-p1-n2-p2...are in the caption. I see the maximum at 500 MHz only for n7p7, n6. Why do the author consider 500 MHz as maximum for the carrier concentration. Moreover QW1 and QW2 do not reach 10^19, only QW7 does. what am I  missing?

Why is so important showing all the carrier from n1p1 to n7p7, this should be clear in the text and/or in the caption.

In Fig8b is Power Efficiency

Fig8d and e: after all the discussion about the thickness of the insulator and the applied voltage frequencies I would expect a conclusion about the trade of between those parameters in order to get a reasonable emission rate and power efficiency.  

R325: it has been already said few rows before and  I do not think this is true.

Round 2

Reviewer 1 Report

The revised manuscript was amended to reflect the reviewer's comment properly. I agree the manuscript is satisfactory to publish in Nanomaterials.

Reviewer 3 Report

I am pleased with the modification.

Best regards